# Water Intake and Adiposity Outcomes among Overweight and Obese Individuals: A Systematic Review and Meta-Analysis of Randomized Controlled Trials

**DOI:** 10.3390/nu16070963

**Published:** 2024-03-27

**Authors:** Qiao-Yi Chen, Jaewon Khil, NaNa Keum

**Affiliations:** 1Department of Food Science and Biotechnology, Dongguk University, Goyang 10326, Republic of Korea; chenqiaoyi0505@naver.com (Q.-Y.C.); kyk3079@naver.com (J.K.); 2Departments of Nutrition, Harvard T.H. Chan School of Public Health, Boston, MA 02115, USA

**Keywords:** water intake, adiposity, systematic review, meta-analysis, randomized controlled trial, weight loss, overweight, obesity

## Abstract

Background: Water consumption is believed to be a key factor in weight management strategies, yet the existing literature on the subject yields inconsistent findings. To systematically assess the scientific evidence regarding the effect of water intake on adiposity, we conducted a systematic review and meta-analysis of randomized controlled trials (RCTs) among overweight and obese populations. Methods: PubMed and Embase were searched for relevant articles published up to December 2023. The summary weighted mean difference (WMD) and 95% confidence interval (CI) were estimated using the DerSimonian–Laird random-effects model. Results: In this meta-analysis of eight RCTs, interventions to promote water intake or to substitute water for other beverages as compared to the control group resulted in a summary WMD of −0.33 kg (95% CI = −1.75–1.08, *I*^2^ = 78%) for body weight, −0.23 kg/m^2^ (95% CI = −0.55–0.09, *I*^2^ = 0%) for body mass index (BMI), and 0.05 cm (95% CI = −1.20–1.30, *I*^2^ = 40%) for waist circumference (WC). Among RCTs substituting water for artificially sweetened beverages, summary WMD was 1.82 kg (95% CI = 0.97–2.67, *I*^2^ = 0%) for body weight and 1.23 cm (95% CI = −0.03–2.48, *I*^2^ = 0%) for WC. Conversely, among RCTs substituting water for sugar-sweetened beverages, summary WMD was −0.81 kg (95% CI = −1.66–0.03, *I*^2^ = 2%) for body weight and −0.96 cm (95% CI = −2.06–0.13, *I*^2^ = 0%) for WC. Conclusions: In conclusion, water intake may not significantly impact adiposity among overweight and obese individuals. However, replacing sugar-sweetened beverages with water might offer a modest benefit in inducing weight loss.

## 1. Introduction

Overweight and obesity are defined as a body mass index (BMI) of 25 to 29.9 kg/m^2^ and ≥30 kg/m^2^, respectively [1]. According to the World Health Organization, as of 2022, 2.5 billion individuals aged 18 years and older, which amounts to approximately 43% of adults worldwide, were classified as overweight or obese [1]. With obesity reaching an epidemic proportion globally, an estimated 1.35 billion adults are expected to become overweight and 573 million obese by 2030 worldwide. It is well established that overweight and obesity are major risk factors for chronic diseases including type 2 diabetes, cardiovascular disease, and cancer compared to normal weight [2,3,4]. Therefore, overweight and obese individuals attempt to lose weight through dietary modifications and exercise. Nevertheless, lifestyle modification is challenging, and only a small fraction of individuals succeed in weight loss through dietary modification, exercise, and behavioral counseling [5]. Consequently, some individuals resort to invasive methods such as diet pills and gastric bypass surgery, which could have potential side effects [6,7]. This underscores the critical need for straightforward and sustainable strategies for weight loss and maintenance.

One of the widely employed strategies is increasing water intake, as it has the potential to reduce energy intake and boost energy expenditure [8,9,10,11]. For instance, drinking water before a meal induces gastric distension, which reduces appetite and increases satiety, subsequently leading to lower energy intake [11,12,13]. Additionally, water intake has been suggested to influence plasma norepinephrine levels, which promote sympathetic activities such as stimulating thermogenesis, thereby increasing energy expenditure [14,15].

Nevertheless, previous randomized controlled trials (RCTs) of overweight and obese individuals showed inconsistent results in terms of the effect of water drinking intervention on weight control, as measured by body weight, body mass index (BMI), and waist circumference (WC) [8,16,17,18,19,20,21,22]. For instance, when participants were instructed to increase water intake or to replace beverages with water, some RCTs reported weight loss effects [8,17,19], while others reported no significant benefit [16,18,21,22]. One RCT even reported an increase in body weight, or WC, among the water intervention group compared to the control group [20]. While scientific evidence on the weight control benefits of water is inconsistent, increasing water consumption, compared to other acclaimed weight loss strategies (e.g., lifestyle modification, supplements, medical procedures), is readily attainable for everyone, affordable, and safe. Therefore, to systematically summarize scientific evidence on the effect of water drinking on inducing changes in adiposity among overweight and obese individuals, we conducted a meta-analysis of RCTs, the study design considered to be the gold standard for making causal inferences in epidemiologic research.

## 2. Methods

The design, analysis, and reporting of this systematic review and meta-analysis were performed in accordance with the Preferred Reporting Items for Systematic Reviews and Meta-Analyses (PRISMA) guideline [23] (Appendix A). Two authors (QYC and JK) participated independently in the database search, study selection, data abstraction, risk of bias assessment, and certainty of evidence assessment. Any discrepancy in the process between the two authors was resolved through discussion with NK.

### 2.1. Study Search

A comprehensive systematic search was conducted in both PubMed and Embase, encompassing articles published up to December 2023. The search terms were structured to include two main components: (1) exposure (i.e., water intake) and (2) outcome (i.e., adiposity). The exposure component comprised the following four keywords: water intake, water consumption, water drink, and drinking water. The outcome component comprised the following twelve keywords: body weight, weight loss, weight gain, weight change, weight control, body mass index, BMI, waist circumference, body composition, obesity, obese, and overweight. For a comprehensive search strategy, we consulted with a librarian, an expert database searcher. The detailed search terms are provided in Appendix A.

The search was limited to human studies and articles written in the English language, and no other restrictions were imposed. The language restriction was applied because the majority of peer-reviewed articles are published in English, and RCTs published in non-English languages are of low quality with methodological limitations [24].

### 2.2. Study Selection

The study selection process adhered to the PICOS (population, intervention, comparison, outcome, study design) criteria outlined in Table 1. To be eligible for inclusion in this systematic review and meta-analysis, studies needed to meet the following criteria: they had to be RCTs investigating the effect of intervening in water intake (addition or substitution) and adiposity, as measured by changes in body weight, BMI, or waist circumference (WC), specifically among overweight and obese participants. In detail, the addition involved promoting water intake, and the substitution recommended replacing other beverages with water. Abstracts and unpublished results were excluded from consideration in order to ensure the inclusion of studies with comprehensive data and rigorous methodologies. When multiple publications existed for the same trial, preference was given to the publication with a follow-up period similar to other included trials [17,20] or the publication that investigated the effects of water intake on weight loss [8] rather than on weight maintenance after initial weight loss [25]. To identify additional relevant papers, we also reviewed reference lists of selected articles, previous systematic reviews, and trial registries. After a rigorous screening process, a total of eight trials were eligible for this meta-analysis.

### 2.3. Data Abstraction

From each RCT, the following information was extracted: first author, publication year, trial name, characteristics of the study population (e.g., country, age, sex), intervention definition (addition vs. substitution), control definition, outcome definition (i.e., body weight, BMI, WC), mean/mean difference and corresponding standard deviation/standard error/95% confidence interval, and variables controlled for (Appendix A).

### 2.4. Risk of Bias Assessment

The risk of bias assessment was based on the revised Cochrane risk-of-bias tool for randomized trials (RoB 2) [26]. The RoB 2 tool has five domains: (1) bias arising from the randomization process; (2) bias caused by deviations from the intended interventions; (3) bias from missing outcome data; (4) bias in the measurement of the outcome; (5) bias in the selection of the reported results. Each study was assessed across these domains, and the overall risk of bias was categorized as low, some concerns, or high risk based on the collective evaluation. The summary of the risk of bias was visually presented using a traffic light plot (Appendix A).

### 2.5. Statistical Analyses

To estimate the effect of the intervention to promote water intake on adiposity (i.e., body weight, BMI, WC), the weighted mean difference (WMD) and 95% CI were calculated using the DerSimonian–Laird random-effects model, which accounts for both between-study variations and within-study variations [27]. Heterogeneity in the relationship between water intake intervention and adiposity across trials was quantified by the *I*^2^ statistic, which indicates the percentage of total variation across studies that is due to between-study heterogeneity [28]. As suggested by Higgins and colleagues, *I*^2^ values of 25%, 50%, and 75% represent low, moderate, and high heterogeneity, respectively [29]. The potential for small study effects, such as publication bias, was assessed using Egger’s test [30]. The test regresses effect estimates on their standard errors on the natural log scale, and the regression intercept of zero is equivalent to the null hypothesis of no publication bias [31].

We performed subgroup analyses and meta-regressions with a priori selected variables (potential effect modifiers). Specifically, we explored heterogeneity in the relationship by the method of interventions used to promote water intake (addition vs. substitution). Additionally, among trials that promoted substituting water for other beverages, we further explored heterogeneity by the type of beverage replaced (artificially sweetened beverages, sugar-sweetened beverages). Of note, subgroup analyses were not performed in cases where the total number of trials was less than three, as the results from a meta-analysis involving a limited number of trials are not informative due to low precision.

In addition, to ensure the robustness of our findings, a sensitivity analysis was performed by excluding a study identified to have a high risk of bias in the risk of bias assessment.

For statistical significance, the two-sided α was set at 0.05. All statistical analyses were conducted using STATA 18 (StataCorp, College Station, TX, USA).

### 2.6. Certainty of Evidence Assessment

To rate the scientific certainty of evidence from our findings, we followed the grading of recommendations, assessment, development, and evaluation (GRADE) approach [32]. The GRADE approach involved systematic consideration of factors such as risk of bias, inconsistency, indirectness, imprecision, and publication bias and categorized evidence into four levels: (1) high, (2) moderate, (3) low, (4) very low.

## 3. Results

### 3.1. Characteristics of Included RCTs

A total of 4314 potential publications were identified using the reported search strategy. Following the removal of duplicates, 3447 publications remained. Screening of titles and abstracts against predefined inclusion and exclusion criteria resulted in a preliminary selection of 43 research articles. Subsequently, upon thorough examination of the full texts of these articles, 35 were deemed ineligible and excluded from our analysis. Finally, a total of eight RCTs contributed to the meta-analyses of water intake with body weight (seven trials), BMI (four trials), and WC (six trials). The study selection process is summarized in Figure 1.

The main characteristics of the included RCTs are summarized in Appendix A. In brief, four RCTs were conducted in the USA [8,20,21,22], two RCTs in Europe [18,19], one in Iran [17], and one in Mexico [16]. Seven of the RCTs targeted overweight and obese adults with a BMI > 25 kg/m² [8,16,17,18,19,20,21], while one trial focused on overweight and obese adolescents with a BMI ≥ 85th percentile [22]. Of these, three RCTs had a follow-up period of three months [8,19,20], while the remaining five extended to six months [16,17,18,21,22]. Regarding the intervention methods used to promote water intake, three RCTs focused on increasing water consumption [8,19,22], whereas the remaining five RCTs recommended substituting water for other beverages [16,17,18,20,21]. Among the five substitution trials, three RCTs advised replacing sugar-sweetened beverages [16,18,21], while the remaining two RCTs advised replacing artificially sweetened beverages [17,20].

A total of eight RCTs were assessed for their risk of bias. Five RCTs [16,19,20,21,22] had a low risk of bias. Due to deviations from the intended interventions and missing outcomes, two RCTs [8,17] raised some concern, and one RCT [18] was determined to have a high risk of bias.

### 3.2. Water Intake and Body Weight

In a meta-analysis of seven trials [16,17,18,19,20,21,22], the summary WMD in body weight comparing the water intervention group vs. the control group was −0.33 kg (95% CI = −1.75–1.08, *p* = 0.64) with high heterogeneity (*I*^2^ = 78%) (Figure 2A). There was no evidence of small study effects, such as publication bias (P_egger_ = 0.44).

The relationship between water intake and body weight was not significantly heterogeneous by the method of intervention to promote water intake (addition vs. substitution) (P_heterogeneity_ = 0.72, Figure 3A). Nevertheless, among the five trials that recommended substituting water for other beverages [16,17,18,20,21], the results varied significantly by the type of beverage replaced (P_heterogeneity_ = 0.02, Figure 4A). Whereas substituting water for artificially sweetened beverages was positively associated with body weight (WMD = 1.82 kg, 95% CI = 0.97–2.67, *I*^2^ = 0%), substituting water for sugar-sweetened beverages was not (WMD = −0.81 kg, 95% CI = −1.66–0.03, *I*^2^ = 2%).

When sensitivity analyses were conducted by excluding the study with a high risk of bias [18], the aforementioned results remained consistent (Appendix A).

### 3.3. Water Intake and BMI

In a meta-analysis of four trials [8,16,17,22], the summary WMD in BMI comparing the water intervention group vs. the control group was −0.23 kg/m^2^ (95% CI = −0.55–0.09, *p* = 0.16) with no evidence of heterogeneity (*I*^2^ = 0%) (Figure 2B). There was no evidence of small study effects, such as publication bias (P_egger_ = 0.61).

The relationship between water intake and BMI did not vary significantly by the method of intervention to promote water intake (addition vs. substitution) (P_heterogeneity_ = 0.99, Figure 3B). Among the two trials that recommended substituting water for other beverages [16,17], heterogeneity by the type of beverage replaced (artificially sweetened beverages vs. sugar-sweetened beverages) was not explored due to the small number of trials.

### 3.4. Water Intake and WC

In a meta-analysis of six trials [8,16,17,20,21,22], the summary WMD in WC comparing the water intervention group vs. the control group was 0.05 cm (95% CI = −1.20–1.30, *p* = 0.94) with moderate heterogeneity (*I*^2^ = 40%) (Figure 2C). There was no evidence of small study effects, such as publication bias (P_egger_ = 0.83).

The relationship between water intake and WC was not significantly heterogeneous by the method of intervention to promote water intake (addition vs. substitution) (P_heterogeneity_ = 0.74, Figure 3C). Among the four trials that recommended substituting water for other beverages [16,17,20,21], the relationship between water intake and WC was not significantly heterogeneous with the type of beverage replaced (P_heterogeneity_ = 0.12, Figure 4B). Notably, although not statistically significant, substituting water for artificially sweetened beverages was suggestively positively associated with WC (WMD = 1.23 cm, 95% CI = −0.03–2.48, *I*^2^ = 0%), whereas substituting water for sugar-sweetened beverages was not (WMD = −0.96 cm, 95% CI = −2.06–0.13, *I*^2^ = 0%).

### 3.5. Certainty of Evidence

According to the GRADE approach, the level of evidence for our findings was assessed as low for body weight, moderate for BMI, and moderate for WC. Certainty was rated down for body weight in the domain of inconsistency due to the absence of overlap in WMD’s 95% CI across the included RCTs, reflecting high heterogeneity (*I*^2^ > 75%). Additionally, in the domain of imprecision, certainty was also downgraded for body weight, BMI, and WC due to 95% CIs, including the null effect (Appendix A).

## 4. Discussion

In this meta-analysis of RCTs conducted among overweight and obese populations, water intake interventions did not have a significant effect on adiposity, as assessed by body weight, BMI, and WC. Moreover, the relationships did not vary significantly according to the method of intervention to promote water intake (addition vs. substitution). Nevertheless, a noteworthy pattern was observed among the trials that recommended beverage substitution. Specifically, when artificially sweetened beverages were substituted with water, there was suggestive evidence of increased adiposity. Conversely, when sugar-sweetened beverages were substituted with water, albeit marginally insignificant, a reduced adiposity was observed.

Water intake has long been hypothesized to aid in weight loss by inducing satiety and reducing energy intake [9,33]. Yet, we found no significant effect of the water intake intervention on adiposity, suggesting the absence of a substantial effect of water on the long-term average energy intake. Indeed, most of the trials included in our meta-analysis reported no difference in energy intake between the intervention and control groups at the end of this study [8,16,18,21,22]. A potential explanation involves caloric compensation, which refers to the regulation of caloric intake by adjusting one’s intake in response to changes in previous consumption [34]. According to this theory, if the feeling of satiety induced by water consumption does not persist until the next mealtime, individuals may be inclined to indulge in snacks between meals, offsetting any potential calorie reduction achieved through water intake [35].

Nevertheless, given our finding that the substitution of sugar-sweetened beverages with water was associated, albeit marginally insignificant, with reduced adiposity, the presence of a beneficial effect of water intake on weight control cannot be ruled out. With the nature of weight regulation being complex and multifactorial, adiposity is influenced by a wide array of factors, including genetic predisposition, dietary habits, and physical activity [36]. If water intake has a true but modest effect on energy intake or metabolism, its effect on weight management may be overshadowed by other influential factors. Furthermore, the intervention periods of the trials in this meta-analysis were relatively short (i.e., 3 months, 6 months). Long-term studies are crucial, especially if the effect of water intake on weight management is small, in order for the effect to accumulate and manifest as changes in adiposity [19].

Alternative explanations relate to the effect of water intake on the sympathetic nervous system. In a previous study of patients with orthostatic hypotension, consumption of 480 mL of water stimulated the sympathetic nervous system, as indicated by elevated plasma norepinephrine levels [37]. Activation of the sympathetic nervous system increases thermogenesis, resulting in an increased energy expenditure. Of note, in overweight and obese individuals who often have a high prevalence of insulin resistance, plasma norepinephrine levels are already elevated [38,39]. As a result, the weight-reducing effect of water intake facilitated by an increase in plasma norepinephrine levels might not manifest due to insulin resistance among overweight and obese individuals.

Unlike our meta-analysis, which focused on overweight and obese individuals, some studies have explored the relationship between water intake and obesity in the general population, which has a low prevalence of insulin resistance. While some studies observed no effect of higher water intake on adiposity outcomes [40,41], there are studies that reported an inverse association [42,43,44,45]. Given the observation of beneficial effects of water intake on weight management in the general population, we cannot rule out the possibility that variations in the effect of water intake on weight management across different demographic groups have different physiological conditions, such as insulin resistance.

In our meta-analysis of the trials that promoted beverage substitution, albeit the number of trials was limited, the conflicting results by the type of beverage replaced were worth discussion. First, among the trials that recommended replacing artificially sweetened beverages with water, increased adiposity was suggested. This finding may likely be due to chance, considering the small number of trials (only two trials), the predominant weight given to one trial [20], the inconsistency in the statistical significance of a positive association across measures of adiposity (body weight and WC), and the known adverse effects of artificially sweetened beverages on gut microbiota diversity [46,47,48] and metabolic regulation [49], both of which are considered etiologic factors of obesity. Nevertheless, the true benefit of artificially sweetened beverages over water on adiposity control cannot be ruled out. A hypothesis has suggested that artificial sweeteners may satisfy people’s craving for sweetness, leading to a reduction in the consumption of other sweetened foods [50].

On the contrary, among the trials that recommended replacing sugar-sweetened beverages with water, a reduction in body weight and WC was suggested. While the validity of the results was limited by the small number of studies, sugar-sweetened beverages are notorious for their high calorie content and added sugars, which have been consistently linked to an increased risk of obesity [51]. Notably, sugar-sweetened beverages are high in fructose [51], which promotes fat accumulation in the liver and de novo lipogenesis, leading to metabolic disturbances and obesity [52,53]. This mechanism was also suggested in one of the RCTs included in our analysis, which showed a significant decrease (*p* = 0.01) in visceral fat, an important indicator of lipogenesis, among individuals consuming water compared to those consuming cola [18]. Furthermore, high consumption of sugar-sweetened beverages that result in a spike in blood glucose and insulin levels is positively associated with the development of glucose intolerance and insulin resistance, all of which are key contributors to obesity [54]. By drinking water in lieu of sugar-sweetened beverages, individuals can avoid these adverse effects associated with added sugars, which, when accumulated, may exert a beneficial effect in reducing adiposity.

Our meta-analysis has several limitations. Firstly, the number of RCTs included in this meta-analysis (eight trials) and the sample size of individual RCTs (ranging from 23 to 276 participants) were relatively small, which limits the statistical power to detect a significant association. Secondly, a high degree of heterogeneity was noted in some meta-analyses, which may limit the interpretability of the summary associations. To address this, we employed the random-effects model as the weighting scheme and explored potential sources of heterogeneity by conducting diverse subgroup analyses using intervention methods. Thirdly, as the included trials were of short duration (3 months, 6 months), our meta-analysis was not able to summarize the long-term effect of water intake on adiposity change. Finally, while it is advised to register protocol before conducting a meta-analysis [55], our meta-analysis was not preregistered. Yet, we adhered to the PRISMA guideline [23], enhancing the transparency and integrity of the study process and reporting.

Yet, there are some strengths in our meta-analysis. To the best of our knowledge, this is the first meta-analysis summarizing the relationship between water intake and adiposity in overweight and obese populations. As our meta-analysis is based on RCTs, the study design with the highest validity in epidemiological studies, our findings are less prone to biases such as confounding, recall bias, and reverse causation. By conducting diverse subgroup analyses according to the method of intervention to promote water intake (addition vs. substitution) and the type of beverages replaced (sugar-sweetened beverages, artificially sweetened beverages), our study aimed to identify an optimal strategy to drink water to control adiposity.

## 5. Conclusions

Among overweight and obese individuals, increased water intake might not have a significant effect on adiposity over a short-term period. However, replacing sugar-sweetened beverages with water might confer a modest benefit against adiposity. More studies are warranted to confirm our findings and better understand the effect of replacing artificially sweetened beverages with water on adiposity.

## Figures and Tables

**Figure 1 nutrients-16-00963-f001:**
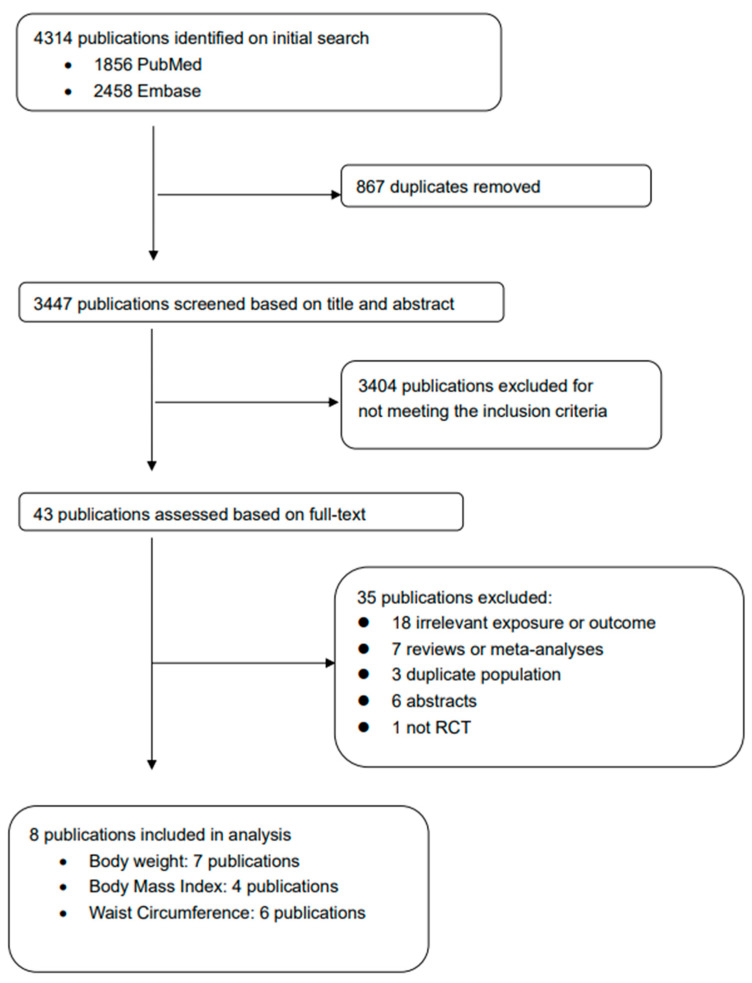
Flowchart for study selection.

**Figure 2 nutrients-16-00963-f002:**
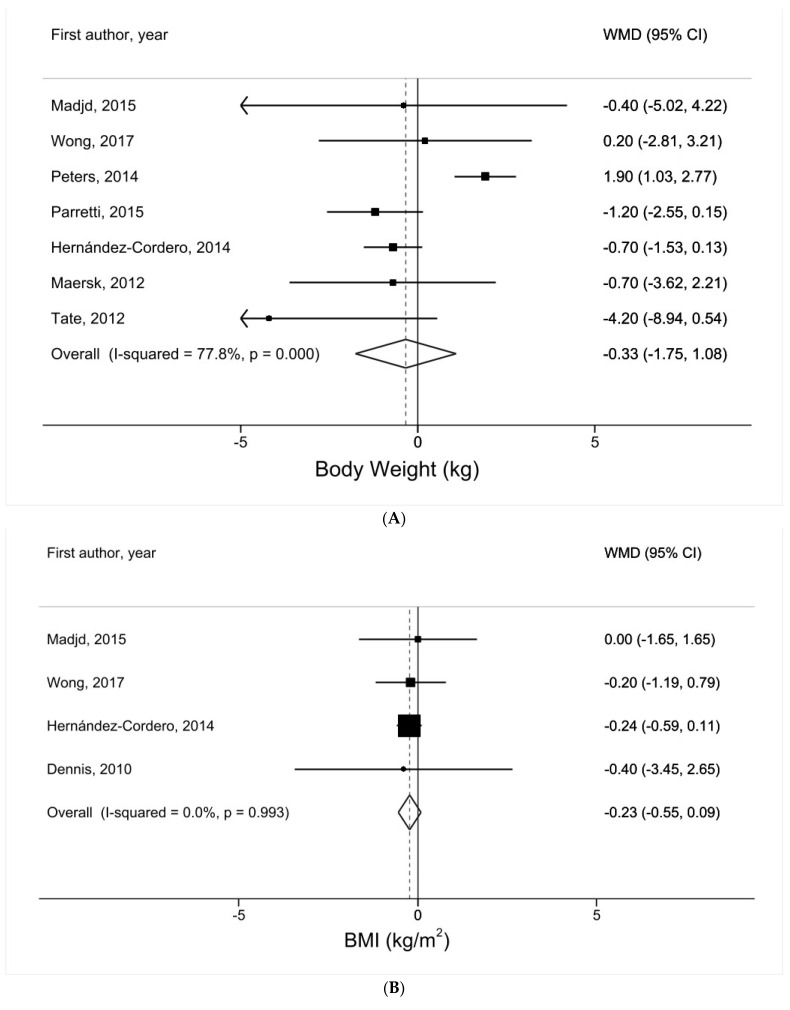
Meta-analysis of adiposity comparing the water intervention group vs. the control group: (**A**) body weight [16,17,18,19,20,21,22]; (**B**) BMI [8,16,17,22]; (**C**) WC [8,16,17,20,21,22].

**Figure 3 nutrients-16-00963-f003:**
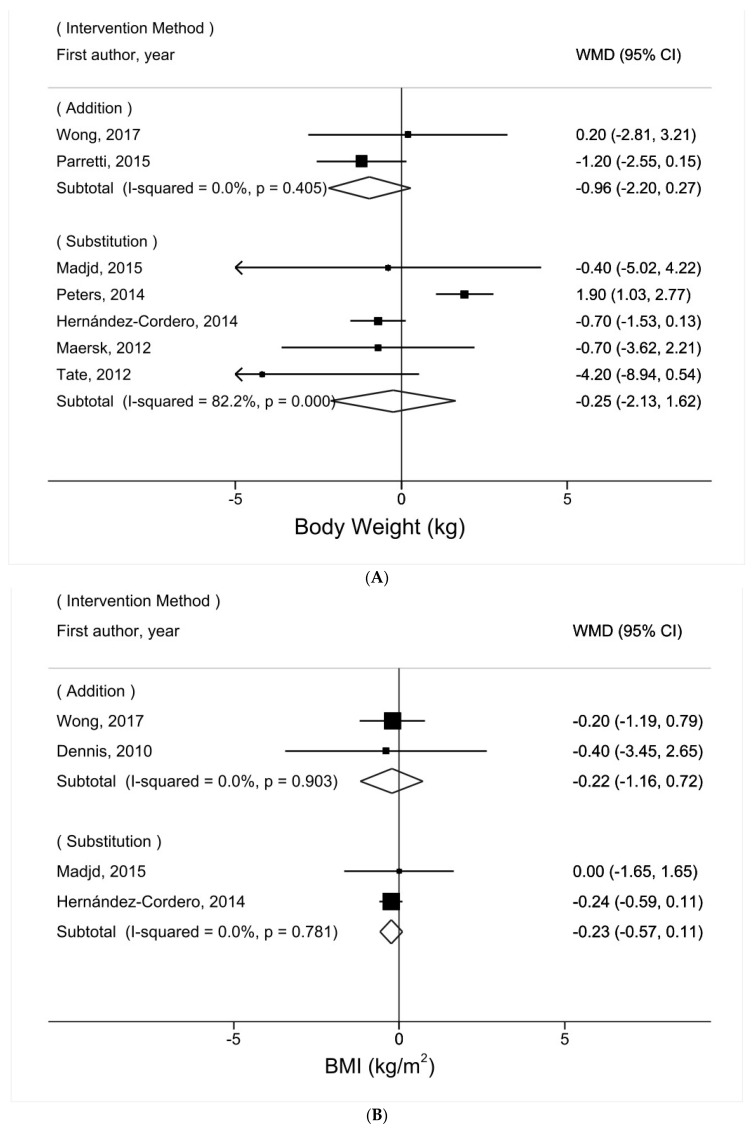
Subgroup meta-analysis of adiposity comparing the water intervention group vs. the control group according to water intervention methods (adding water, substituting water for other beverages): (**A**) body weight [16,17,18,19,20,21,22]; (**B**) BMI [8,16,17,22]; (**C**) WC [8,16,17,20,21,22].

**Figure 4 nutrients-16-00963-f004:**
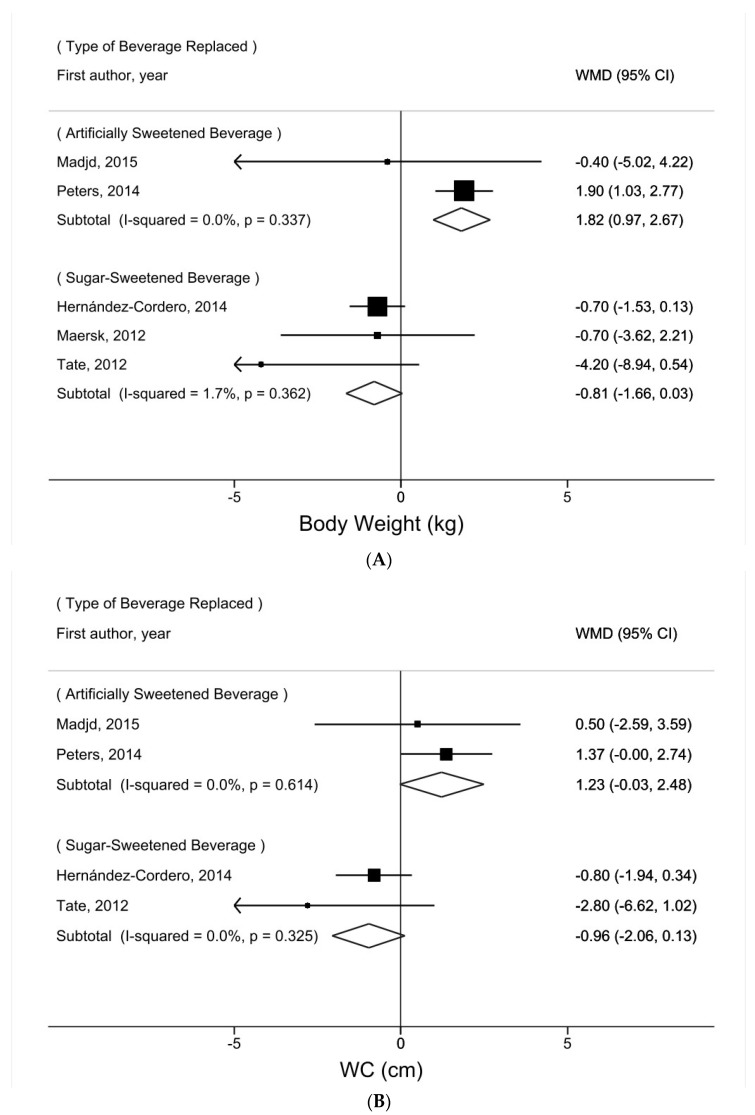
Subgroup meta-analysis of adiposity comparing the water intervention group vs. the control group according to the types of beverages replaced (artificially sweetened beverage, sugar-sweetened beverage): (**A**) body weight [16,17,18,20,21]; (**B**) WC [16,17,20,21].

**Table 1 nutrients-16-00963-t001:** PICOS criteria for inclusion of RCTs.

Parameter	Inclusion Criteria
Population	Overweight and/or obese individuals
Intervention	Water intake intervention
Comparator	Any other beverages, placebo
Outcome	Body weight, body mass index, waist circumference
Study design	Randomized controlled trials

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
