# Peer review of "Water Intake and Adiposity Outcomes among Overweight and Obese Individuals: A Systematic Review and Meta-Analysis of Randomized Controlled Trials"

_nutrients, 2024, doi:10.3390/nu16070963_

Round 1
Reviewer 1 Report
Comments and Suggestions for Authors
There has been much speculation for some years suggesting that an increased water intake assists with weight control. This paper analyses the available data obtained by carrying out a systematic review and meta-analysis of randomized controlled trials. The findings are therefore likely to be of much interest.
I assume that the search method used can be categorized as a systematic review. In that case the term “systematic review” should be included in the title, abstract and Methods.
Keywords, delete “meta-analysis” and “randomized controlled trial”, add “weight loss”
Line 61, were there language restrictions on the articles selected? Many systematic reviews only use papers published in English.
Line 102, I assume there were two RCTs in Europe (not “wo”)
In the Abstract and the Results section of the paper and in Figures 2 and 3 it is important to add units to the different measures. It also important to make it clear in both the text and figures whether drinking more water (rather than other beverages) leads to higher weight, BMI and MC or a decrease.
Lines 146-148, this sentence is misleading as it seems to suggest that you are referring to findings from an observational study, such as a cohort study. The same problem also occurs in the next paragraph (“we observed no association between water intake and adiposity”).
There are numerous places where some improvements can be made to the quality of the writing. I encourage the authors to have the paper edited by someone skilled in scientific English.
Reviewer 2 Report
Comments and Suggestions for Authors
-
The authors have omitted the prepublication of a protocol in a protocol repository such as PROSPERO or OSF, raising concerns regarding transparency and reliability.
-
The authors are encouraged to adhere to Cochrane guidelines regarding the search in electronic databases, ensuring alignment with a systematic review process.
-
It is recommended that the authors utilize the RoB 2.0 tool to evaluate the risk of bias in the included studies.
-
Please incorporate details in the statistical analysis section regarding the model used for conducting the meta-analysis, the heterogeneity estimator employed, and the approach taken for handling data not reported as mean and standard deviation.
-
Additionally, the authors are urged to include prediction intervals alongside the 95% confidence intervals.
-
The authors should also conduct the GRADE approach to assess the certainty of their findings.
-
Figure 1 requires modification to address missing information.
-
Table S3 should be revised to rectify discrepancies, particularly in the column where the trial name is expected, as the current entry "weight loss intervention" does not align with the column title.
Comments on the Quality of English Language
Minor editing of English language required
Reviewer 3 Report
Comments and Suggestions for Authors
In general terms, the article is well written, easy to understand and relatively extensive. I think it is an interesting topic and I encourage the authors to continue this line of research. Some aspects need to be improved and are indicated below.
Keywords
It would be recommended to include overweight and obesity as keywords, in case the number does not exceed the norm established by the journal.
Introduction
I consider that the introduction is well written and contains the main ideas for understanding the study. However, I think that the authors should expand a little more on certain aspects such as the definition of overweight and obesity and the current epidemiological figures on incidence, prevalence, among others.
As for the concepts of energy expenditure, it would be useful to explain in more detail the mechanisms of action that have been included in line 36-38. Explaining it in more detail would facilitate the reader's understanding.
In addition, it would be interesting if the authors could briefly justify why they chose this type of study. Why did they analyse water consumption? What were their reasons for undertaking this study? It seems to me to be an original study and I think that clarifying this would add value to the article.
Material and methods
In section 2.1. Study search, the authors indicate that articles in English were selected, why was the search not extended to other articles written in a different language?
In section 2.2. Study selection, the authors indicate that BMI, WC and body weight were assessed. Did the authors find any articles that took into account the visceral fat index of overweight or obese patients?
Results
The results are clearly explained, easy to understand and reflect the main findings of the study. I consider that no modifications are necessary in this section.
Discussion
Lines 184 and 185 indicate that, according to some studies, the consumption of sweetened beverages may decrease the craving for sweets, leading to a decrease in fat. However, have these studies analysed the effect of sweeteners on the gut microbiota? This would be interesting because gut microbiota is also considered as an etiology of obesity. In my opinion, analysing only the effect of something in terms of weight would not be entirely correct, as the success of treatment of an overweight or obese patient is not only measured by weight loss. There are other markers (both anthropometric and non-anthropometric) that are also indicative of treatment success.
In line 195, the authors could complement the information by indicating that the high levels of simple sugars contained in this type of sugar-sweetened beverages also influence the patient's glycaemic control and increase the risk of insulin resistance characteristic of patients with obesity.
In line 215 the authors conclude that the intake of sugar-sweetened beverages with water might confer modest benefit against adiposity.
However, do these studies indicate negative or detrimental effects, and do they only look at the effect on adiposity? As mentioned above, sweetener consumption can damage the gut microbiota. I believe that the authors should also indicate the negative effects of sweetener consumption in order to get a broader picture of the problem that would make it easier to contrast the positive and negative aspects.
The conclusion of the study is embedded in the discussion. Where journal rules allow, I consider it more appropriate to separate the discussion from the conclusion.
Round 2
Reviewer 2 Report
Comments and Suggestions for Authors
1) In the limitations section of their manuscript, the authors should underscore that the absence of a pre-published protocol may potentially affect the reliability of their findings.
2) All systematic reviews should have a comprehensive literature search strategy including:
2.1) search at least 2 databases
2.2) provide keywords and the search strategy for at least one database
2.3)justify publication restrictions
2.4) search the reference list of the included studies
2.5) search trial/study registries
2.6) consult content experts in the field
2.7) search for grey literature
Comments on the Quality of English LanguageMinor editing of English language required
